# Effect on Chest Compression Fraction of Continuous Manual Compressions with Asynchronous Ventilations Using an i-gel^®^ versus 30:2 Approach during Simulated Out-of-Hospital Cardiac Arrest: Protocol for a Manikin Multicenter Randomized Controlled Trial

**DOI:** 10.3390/healthcare9030354

**Published:** 2021-03-20

**Authors:** Loric Stuby, Laurent Jampen, Julien Sierro, Erik Paus, Thierry Spichiger, Laurent Suppan, David Thurre

**Affiliations:** 1Genève TEAM Ambulances, Emergency Medical Services, CH-1201 Geneva, Switzerland; d.thurre@gt-ambulances.ch; 2ESAMB—École Supérieure de Soins Ambulanciers, College of Higher Education in Ambulance Care, CH-1231 Conches, Switzerland; laurent.jampen@edu.ge.ch; 3Compagnie d’Ambulances de l’Hôpital du Valais, Emergency Medical Services, CH-1920 Martigny, Switzerland; julien.sierro@hopitalvs.ch; 4SPSL—Service de Protection et Sauvetage Lausanne, Emergency Medical Services, CH-1005 Lausanne, Switzerland; Erik.Paus@spsl-lausanne.ch; 5ES ASUR, Vocational Training College for Registered Paramedics and Emergency Care, CH-1052 Le Mont-sur-Lausanne, Switzerland; t.spichiger@es-asur.ch; 6Ambulance Riviera, Association Sécurité Riviera, Emergency Medical Services, CH-1814 La Tour-de-Peilz, Switzerland; 7Division of Emergency Medicine, Department of Anesthesiology, Clinical Pharmacology, Intensive Care and Emergency Medicine, University of Geneva Hospitals and Faculty of Medicine, CH-1211 Geneva, Switzerland; laurent.suppan@hcuge.ch

**Keywords:** emergency medical services, paramedics, airway, supraglottic airway device, cardiac arrest, i-gel^®^, CPR, prehospital, resuscitation, chest compression fraction

## Abstract

The optimal airway management strategy during cardiopulmonary resuscitation is uncertain. In the case of out-of-hospital cardiac arrest, a high chest compression fraction is paramount to obtain the return of spontaneous circulation and improve survival and neurological outcomes. To improve this fraction, providing continuous chest compressions should be more effective than using the conventional 30:2 ratio. Airway management should, however, be adapted, since face-mask ventilation can hardly be carried out while continuous compressions are administered. The early insertion of a supraglottic device could therefore improve the chest compression fraction by allowing ventilation while maintaining compressions. This is a protocol for a multicenter, parallel, randomized simulation study. Depending on randomization, each team made up of paramedics and emergency medical technicians will manage the 10-min scenario according either to the standard approach (30 compressions with two face-mask ventilations) or to the experimental approach (continuous manual compressions with early insertion of an i-gel^®^ supraglottic device to deliver asynchronous ventilations). The primary outcome will be the chest compression fraction during the first two minutes of cardiopulmonary resuscitation. Secondary outcomes will be chest compression fraction (per cycle and overall), compressions and ventilations quality, time to first shock and to first ventilation, user satisfaction, and providers’ self-assessed cognitive load.

## 1. Introduction

### 1.1. Background

Achieving high-quality cardiopulmonary resuscitation (CPR) requires the provision of chest compressions of adequate depth and rate while avoiding interruptions [1]. The chest compression fraction (CCF) is the proportion of time spent performing compressions, and can be increased by minimizing interruptions. The latest European Resuscitation Council (ERC) recommendations emphasize the importance of applying the highest possible quality of chest compressions with minimal interruption [2], and the American Heart Association (AHA) recommendations state that the CCF should be equal to at least 60%, and ideally exceed 80% [1]. In animal studies, continuous compressions are associated with a higher rate of good neurological outcomes compared to a 30:2 regimen [3,4]. Human clinical trials have shown that maintaining a high CCF is linked to a higher rate of return of spontaneous circulation (ROSC), survival, and favourable neurological outcomes both in shockable and non-shockable rhythms [5,6,7,8,9,10,11]. For instance, Christenson et al. found that a 10% increase in CCF is approximately equal to an 11% increase in survival [8].

The optimal airway management strategy during CPR and after ROSC is uncertain [12,13,14,15]. Endotracheal intubation (ETI) remains the most effective and safest technique, but requires great skill that must be maintained by regular practice to be efficient and avoid complications [16,17,18]. The use of supraglottic airway (SGA) devices during CPR is associated with a lower incidence of regurgitation when compared to face mask ventilation (FMV) [19]. The i-gel^®^ is an effective SGA device that can be rapidly inserted with high success rates [20,21,22,23,24,25,26,27,28,29]. Most of the time, the i-gel^®^, which has a thermoplastic non-inflatable cuff, enables the delivery of continuous chest compressions with providers reporting no leak in the vast majority of patients [30]. This device also confers relative airway protection, as aspiration is less frequent compared to FMV [31]. In the AIRWAYS-2 trial, regurgitation and aspiration rates were not significantly different between the ETI and SGA groups [32].

In addition, the use of SGA enables the adjunction of end-tidal CO_2_ to monitor the quality of CPR and its optimization, which could also be useful to help for ROSC detection [1].

The hypothesis is that the experimental approach could improve CCF during out-of-hospital cardiac arrest (OHCA), with a reasonable time of about 60 s to first effective ventilation.

### 1.2. Objectives

The primary aim of this study is to determine whether the immediate insertion of an i-gel^®^ while providing continuous chest compressions followed by asynchronous ventilations can increase the CCF compared to the 30 compressions/two ventilations standard approach in a simulated model of OHCA after a short teaching intervention.

The secondary objective is to compare the basic life-support (BLS) quality by assessing the chest compressions and ventilations parameters using the International Liaison Committee on Resuscitation (ILCOR), ERC and AHA up-to-date recommendations [1,33,34].

## 2. Materials and Methods

### 2.1. Study Design

This will be a multicenter, parallel, randomized, superiority simulation study. This protocol was designed in accordance with the Standard Protocol Items: Recommendations for Interventional Trials (SPIRIT) statement (SPIRIT Checklist as Appendix A) [35]. The Consolidated Standards of Reporting Trials (CONSORT) flow chart of the trial is displayed in Figure 1 and the SPIRIT Figure describing the timeline is displayed in Table 1, as recommended [35,36]. 

### 2.2. Setting

In Switzerland, all paramedics graduate from Colleges of Higher Education in Ambulance Care after a 3-year (5400 h) training curriculum covering most aspects of prehospital emergency care. Paramedics are trained to treat most of the prehospital emergencies autonomously and can administer several advanced life-support (ALS) treatments by virtue of emergency care protocols elaborated by prehospital medical directors. Emergency medical technicians (EMTs) graduate after a 1-year curriculum focusing on BLS aspects and are able to assist paramedics in taking care of critically ill patients; they can also care for patients with non-life-threatening conditions under direct supervision by a paramedic. 

Due to the Swiss federal system and owing to the individual design of treatment protocols by different medical directors, there is substantial inter-cantonal heterogeneity among Swiss Emergency Medical Services (EMS) and more largely between the healthcare systems in this country. However, the three-tiered response to OHCA is largely similar. The first tier is made of a system of lay or professional first-responders [37], who are registered volunteers dispatched by the emergency medical communication center (EMMC) to perform BLS manoeuvres when OHCA is identified on the call. Simultaneously, the second tier, which is composed of an ALS ambulance staffed either by an EMT and a paramedic or by two paramedics, is also dispatched. The third tier, which consists of a medical reinforcement, is available in most cantonal systems [38].

### 2.3. Trial Centers

The i-gel^®^ device has been largely used in Geneva for years, but some regions, such as Vaud and Valais, have not yet adopted its use widely. Therefore, this study will be performed in four different EMS from the cantons of Vaud and Valais in the French-speaking part of Switzerland. The study centers were the following: *Ambulance Riviera, Association Sécurité Riviera*, La Tour-de-Peilz, Vaud; *Compagnie d’Ambulances de l’Hôpital du Valais*, Martigny, Valais; *Service de Protection et de Sauvetage Lausanne*, Lausanne, Vaud *and Société de Transport en Ambulance Régionale, STAR Ambulances*, Épalinges, Vaud. All these services do not currently use the i-gel^®^ device in the field. They are applying the standard 30:2 approach. 

### 2.4. Participants, Study Flow and Recruitment

All registered paramedics and EMTs actively working in any of the participating study centers will be eligible for inclusion in the study. The exclusion criteria will be the following: member of study investigators, not to have undergone the 20 min workshop, or not to have followed the self-training session supported by the demonstration video. Participants will be recruited by the local study coordinator with the same email template (Appendix A). Participants will be told the study is about OHCA management, but will not be informed of the outcomes studied. 

### 2.5. Study Sequence

The study sequence is displayed in Figure 2.

#### 2.5.1. Randomization

We will use two levels of randomization. Teams will be generated using an online balanced team generator [39]. For each trial center, a stratification by professional status will be made in order to be representative of the current practice in the field with at least one paramedic per team. In teams made up of an EMT and a paramedic, the paramedic will always be the “team leader”. In teams made up of two paramedics, they will choose their role, as in actual clinical practice.

The teams will then be randomized in one of the two study arms by opening an opaque, sealed envelope containing the approach to apply (standard or experimental) created using a stratified randomization list generated online by Loric Stuby (LSt) using a 1:1 ratio [40]. The actual standard 30:2 approach may be slightly different between the EMSs, for instance regarding service-specific task distribution, quality processes, prior specific training or local procedures during CPR. To take into account such discrepancies, a randomization stratified by service will be used.

#### 2.5.2. Pre-Scenario Standardized Workshop

As an introduction, information about the study will be given and participants’ questions will be answered. Written consent will be obtained by LSt and the demographics data questionnaire will be individually completed. 

Then, the use of the i-gel^®^ device use will be taught by one of the investigators (LSt) in accordance with Peyton’s approach [41,42], because this approach has shown superiority [43], and is based on a standard operating procedure (Appendix A) created by study investigators following the manufacturer’s instructions:

(1) The instructor performs a complete insertion sequence in real-time without any comments;

(2) The instructor performs an insertion sequence with step-by-step explanations (description of key points);

(3) The learners guide the instructor step by step to perform the insertion;

(4) The learners do the complete insertion sequence then feedback could be given. They will perform this step a maximum of 3 times.

This workshop will last 20 min.

#### 2.5.3. Self-Managed Training Session

After the workshop, each team will have 20 min to self-train the introduction of the device into the complete OHCA management sequence on a CPR manikin permitting airway management. They will be supported by a demonstration video (https://www.youtube.com/watch?reload=9&v=UuX2cik_aBg, accessed on 23 February 2021), which they can use freely.

#### 2.5.4. Resuscitation Scenario

When entering the study room, an overview of the characteristics of the simulation manikin and of the use of the defibrillator will be given through a standardized video (https://www.youtube.com/watch?v=ifUj8xFSfy0, accessed on 23 February 2021). The team will then be asked to perform a 10 min highly realistic adult CPR scenario on a high-fidelity WiFi manikin. The procedure will be standardized across all sites to ensure that each participant is exposed to exactly the same scenario, with similar challenges in decision-making and treatment provided on the same manikin. The uniform delivery of the scenario will minimize confounders. The room will be exclusively devoted to the simulation to prevent unexpected interruptions or external stimuli. Participants will be told that the resuscitation scenario will be stopped after 10 min, independently of their actions, and that no feedback will be given. The simulation will be carried out without external pressure (witnesses, relatives, road traffic sounds, etc.).

The scenario will start with a clinical statement to recognize the life-threatening condition of the patient, given by one of the investigators as follows: *«Here is Michael, a 50-year-old man who suddenly collapsed 10 min ago. He is now unconscious, pale and seems not to be breathing. The medical reinforcement is already underway and will be on site in about ten minutes. No first responder has been dispatched by the EMMC, and no bystander is on scene».* The team leader will be asked to reformulate this statement (closed-loop communication) to ensure comprehension. He will then choose and open one of the envelopes of the stack. The opaque, sealed envelope will contain the approach they will have to apply: current practice approach or experimental approach. From this point on, there will be no more contact between the participants and the study team, except to stop the scenario.

The simulated patient will be apneic and pulse will be absent if checked. The first compression will be defined as T0. After placement of the pads, the defibrillator’s display will show ventricular fibrillation (VF). To increase simulation’s fidelity CPR waves will be automatically displayed when compressions are delivered. All subsequent rhythm analyses will show refractory VF, regardless of whether a shock is delivered. Participants will be able to obtain an intravenous access on first attempt. They should administer first 1 mg of epinephrine (at the earliest after the second shock), then a first-dose of amiodarone or lidocaine following local protocols (in accordance with the 2020 AHA adult cardiac arrest algorithm) [44]. The scenario will be stopped exactly 10 min after the first compression. There will be no further intervention or educational adjunct after the study period.

#### 2.5.5. Study Groups

The control group will be told to apply their standard of care (30:2 current practice approach). Teams will have access to their usual resuscitation equipment (e.g., they will probably use an oropharyngeal cannula). The decision of whether to use any specific item will remain at the discretion of the team members, as in an actual resuscitation.

We developed an experimental approach which consists, when starting CPR in an OHCA, of delivering no prior FMV, but instead directly placing an i-gel^®^ device while the second team member performs continuous chest compression from the beginning of resuscitation manoeuvres. As soon as the device is in place, ventilations are given asynchronously at a rate of 10 per minute following AHA recommendations [1]. Paramedics and EMTs will be provided with an i-gel^®^ device size 4 (Intersurgical Ltd., Wokingham, UK) and told to apply the experimental approach. A lubricant recommended by the manikin manufacturer will be provided.

#### 2.5.6. Data Collection

Two questionnaires (Appendix A) will be used. The demographics data questionnaire will be filled pre-scenario and will be used to gather participants’ characteristics (age, gender, years of professional experience, professional title, estimated number of i-gel^®^ insertions performed on manikin during last year, estimated number of i-gel^®^ insertion performed on real patient during the last year).

The first part of the assessment’s questionnaire will be used by the two investigators responsible for simulation during the resuscitation scenario to collect the number of attempts needed to reach successful insertion; it will be countersigned by the team leader post-scenario to limit errors (double check). All other CPR outcomes (chest compressions, shocks and ventilations) are measured by the SimMan.

The second part of the assessment’s questionnaire will be completed post-scenario by each participant to assess satisfaction using a 5-point Likert scale ranging from “Not satisfied at all” to “Very satisfied”, and cognitive load using a 9-point symmetrical category scale ranging from “Very, very low mental effort” (1) to “Very, very high mental effort” (9), as described by Paas and Van Merriënboer and previously used in simulation sessions wherein participants have to translate the perceived amount of total mental effort into a numerical value [45,46].

### 2.6. Equipment

For each study session, the same human patient simulator and the dedicated multiparametric monitor/defibrillator (Laerdal SimMan 3G, Laerdal Medical, Stavanger, Norway) will be used. All other equipment required for the scenario will be made available in the usual services intervention bags (factice medications will be used, e.g., sterile water). Teams will be briefed about the location and characteristics of equipment prior to the commencement of the scenario, before randomization. An i-gel^®^ device size 4 (Intersurgical Ltd., Wokingham, UK) and a lubricant tube will be available in the intervention bag.

### 2.7. Outcomes

The primary outcome will be the chest compression fraction (CCF) during the first two minutes of CPR (starting from first compression).

Secondary outcomes will be: CCF of each following 2 min cycle (from minutes 2 to 4, 4 to 6, 6 to 8 and 8 to 10), overall CCF (entire 10 min scenario), chest compressions depth; proportions of compressions within the depth target (5 to 6 cm), below target value (<5 cm) and above target value (>6 cm); chest compressions rate; proportions of compressions within the appropriate rate target (100 to 120 compressions per minute (cpm)), below target value (<100 cpm) and above target value (>120 cpm); proportions of compressions with complete chest recoil (<5 mm deviation from the reference value); time to first shock; time to first effective ventilation (defined as >300 mL [47,48,49,50,51]); ventilations volume; proportions of ventilations above target value (>700 mL), below target (<300 mL) and within target (300–700 mL); number of delivered ventilations; user satisfaction, and providers’ self-assessed cognitive load.

### 2.8. Blinding and Bias Minimization

To minimize bias during the study session, allocation will be disclosed as late as possible. Teams will be allocated only after the manikin and defibrillator characteristics presentation, and after the simulated patient’s condition has been given. From the moment of the allocation, there will be no more contact between investigators and participants. Thus, during the pre-scenario sessions, the teams will not have been allocated to one of the study arms, so the investigator responsible for the device workshop session is blinded to allocation. The participants are therefore also blinded during the workshop as well as in the self-managed training session.

Due to study design, which includes the use of a specific medical device, we will not be able to blind the participants during the study scenario. However, they will be blinded to the outcomes. Assessment bias will be limited, as the outcomes will be collected by an automated Laerdal manikin. The data analyst will also be blinded as to group allocation. Participants will be asked to withhold information until the end of the data collection process.

### 2.9. Sample Size Calculation, Data Extraction and Statistical Analysis

The sample size is based on an estimate as there are few relevant data available to compute it for such a study. The mean time required to deliver two ventilations is estimated to be around 4 s; two ventilations should be provided 5 times during the first cycle, therefore amounting to a total of 20 s of no-flow. In addition, the initial rhythm analysis should take about 8 s. Pooled together, these no-flow times should amount to 28 s out of the first 120 s cycle, therefore representing 23% of no-flow. The CCF should therefore be around 77% in the control group. In the experimental group, taking into account the time for rhythm analysis and shock, estimated to be around 8 s (same time as control group), the corresponding no-flow time should be about 7%, with a CCF of 93%. Based on observational data from case reviews and pilot tests, the variability of the data is estimated with a standard deviation of 12.

Twenty-four teams will be required in order to have a 90% chance of detecting, at the 5% significance level, an increase in the primary outcome measure from 77 in the control group to 93 in the experimental group.

The data extraction process will take place as follows: the CPREvents.xml file will be obtained by unzipping the SSX file proposed by Laerdal after each simulation with the SimMan manikin and saved in a comma-separated values (CSV) file. Then, this file will be uploaded in an SQL table (by phpMyAdmin, https://www.phpmyadmin.net/, V. 5.0.4, accessed on 23 February 2021). The variables of interest will be automatically generated in an accurate, reliable and reproducible manner using a PHP script specifically designed for the study (Appendix A), from which a new CSV file will be extracted and imported directly into the statistics software. To minimize copying or typing errors, all data collected on paper CRF will be entered in duplicate using EpiData [52]. The consistency of the answers will then be checked by merging the two files in a common variable where discrepancies are listed. After resolving the discrepancies, data will be exported in a Stata DTA file. These two Stata databases will then be concatenated. Missing data will be treated as such. No imputation technique will be used.

All data that could allow the data analyst to identify the group allocation will be deleted (e.g., number of insertion attempts). The groups will be renamed otherwise (groups “Teysachaux” and “Moléson”) and the curated database will be sent in Stata DTA file format to DT for formal analysis. All investigators will be able to access the curated and coded data set.

Distribution normality will be checked graphically for continuous variables, and the Shapiro–Wilk test will be used in case of doubt. If this assumption is checked, a Student’s t-test will be used to compare both groups; if not, the Mann–Whitney U test will be used. These variables will be described using mean (SD) or median (Q1; Q3) depending on their distribution. We will test proportions using Fisher’s exact test. They will be reported with their 95% confidence interval.

User satisfaction will be assessed graphically, then dichotomized into “Satisfied” versus “Not satisfied” before being tested using Fisher’s exact test. The cognitive load variable will be treated as continuous.

A two-sided p-value of 0.05 will be considered significant. All the statistical analyses will be performed using Stata V15.1 (StataCorp. 2017. Stata Statistical Software: Release 15. College Station, TX: StataCorp LLC).

## 3. Results, Ethics and Dissemination

### 3.1. Research Ethics Approval

The study has been submitted to the regional Ethics Committee (REQ-2020-01491), which stated that the need for approval was waived, as this study does not come into the scope of the Swiss federal law on human research [53]. The study will be conducted in accordance to the principles of the Declaration of Helsinki and Good Clinical Practice guidelines [54,55]. All participants will sign an informed consent form before randomization (Appendix A).

### 3.2. Protocol Version

The actual protocol version is 1.0 (22 February 2021). Study sessions are scheduled for March and May 2021.

### 3.3. Trial Registration

This study is registered on https://clinicaltrials.gov/ (accessed on 23 February 2021) under trial registration number NCT04736446.

### 3.4. Data Curation and Availability

The results, either positive or negative, will be submitted for publication in a peer-reviewed journal following the CONSORT statement [36]. The curated database will be deposited on Mendeley Data [56]. The original documents and files will be kept in a safe place. One investigator (DT) will be responsible for data and files storage from all centers for 10 years.

### 3.5. Consent or Assent

Information regarding the study and its objectives (without detailed outcomes), including data security, will be provided. Written consent will be gathered. All the documents will be written in French (Appendix A and Appendix A). Participation will be free and each participant will be able to withdraw at any time without giving any justification. The benefit obtained by the participants will be an increase in knowledge of the i-gel^®^ device’s use, which could be useful in their practice. There will be no financial compensation.

## 4. Discussion

The main limitation of this study is that it will be performed on a manikin, and that there will therefore be no direct evidence regarding actual clinical outcomes. Nevertheless, it is reasonable to assume that, if the CCF can be improved on a manikin, the same should hold true in an actual OHCA situation. However, the main difference between simulated and actual OHCA could rely on device insertion rate. Thus, should increases in the CCF be proven in this study, a further step could be a trial conduct in the field.

Due to the type of intervention, the blinding of participants to their allocated group will not be possible. Participants will, however, remain unaware of the study outcomes. Moreover, an objective measure of the main outcomes will be used continuously during the intervention to allow reliable assessment. The voluntary recruitment strategy could bias the sample, via a self-selection phenomenon which would give an unrepresentative sample of participants (with only those who are the most confident, motivated and involved that will take part in the study).

The seal pressure appears to improve over time in a number of humans due to the thermoplastic properties of the gel cuff [27], which may form a more efficient seal around the larynx after warming to body temperature. In this study, we will not expect such a good sealing because the manikin will not reach the appropriate temperature. This could make the manikin more difficult to ventilate.

The scenario represents one of the various contexts of OHCA: for instance, no first responder or medical reinforcement will be present on scene, and therefore the cardiopulmonary resuscitation will be managed with only two healthcare providers. This could impact several aspects of the secondary outcomes, due to fatigue or specific organization, and therefore limit the generalization of the results.

High-fidelity manikins provide a realistic and high-yield immersive environment. They have been shown to induce an increase in physiological and psychological stress parameters [57,58] during emergency training sessions [59] without affecting the quality of cardiopulmonary resuscitation [60]. In addition, the experimental approach will be new to the participants, thereby further increasing their stress level. Therefore, adding other external stressors should not significantly affect the outcomes of this study, but might impede learning. Indeed, new learners initially benefit from a less stressful simulation environment before being confronted with a gradual increase in stress demands through clinical practice [61], or by virtue of progressive simulations [62]. Moreover, should further studies prove the addition of external stressors to be a confounding factor, the random allocation of participants should markedly dampen any effect this factor might have.

The strengths of this trial are the multicenter design, including paramedics and EMTs from several EMS of Switzerland, which, by comparing two feasible and clinically relevant treatment strategies, addresses an evidence gap regarding the optimal approach, especially in the initial airway management of OHCA, developed in accordance with the SPIRIT guidelines. The results of the study could support the external validity and implementation of this experimental approach in the field.

## 5. Conclusions

This study should help to determine whether early use of an i-gel^®^ with continuous manual compressions could increase CCF compared to the 30:2 approach.

## Figures and Tables

**Figure 1 healthcare-09-00354-f001:**
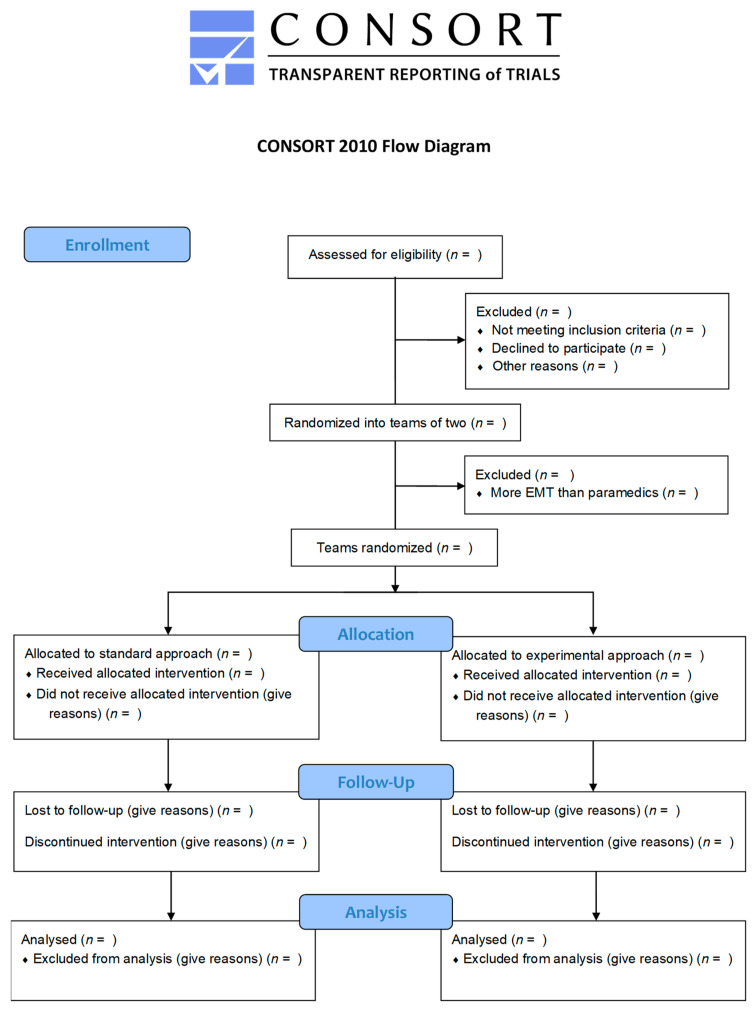
Study flow chart.

**Figure 2 healthcare-09-00354-f002:**
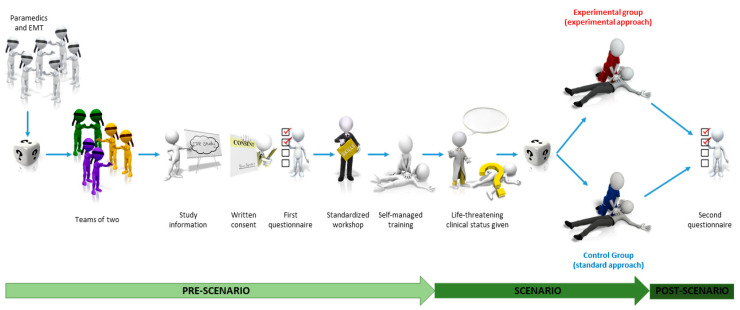
Study sequence.

**Table 1 healthcare-09-00354-t001:** Timeline of enrolment, interventions and assessments.

	STUDY PERIOD					
	Enrolment	Workshop	Training Session	Allocation	Post-Allocation	Close-Out
TIMEPOINT	t-3 _before study day_	t-2 _-40 min_	t-1 _-20 min_	t0	t1	t2
**ENROLMENT:**						
**Eligibility screen**	X	X				
**Randomization into pairs**	X					
**Informed consent**		X				
**Standardized workshop**		X				
**Training session**			X			
**Allocation**				X		
**INTERVENTIONS:**						
**Standard approach**					X	
**Experimental approach**					X	
**ASSESSMENTS:**						
***Age, gender, years of experience, profession title, estimated insertions on manikin, estimated insertions on real patients***		X				
***CCF, compressions depth and rate, chest recoil, time to first shock and to first ventilation, number and volume of ventilations.***					X	
***User satisfaction, cognitive load***						X

## Data Availability

No new data were created or analyzed. Data sharing is not applicable to this article. Data that will be generated in this coming study will be freely available on Mendeley Data.

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
