# Peer review of "Effect on Chest Compression Fraction of Continuous Manual Compressions with Asynchronous Ventilations Using an i-gel® versus 30:2 Approach during Simulated Out-of-Hospital Cardiac Arrest: Protocol for a Manikin Multicenter Randomized Controlled Trial"

_healthcare, 2021, doi:10.3390/healthcare9030354_

Round 1

Reviewer 1 Report

The study is an interesting and elegant experimental model. This is a manikin simulation of the out-of-hospital cardiac arrest scenario that evaluates very specific endpoints of the ACLS procedure, mainly whether the use of a i-gel supraglottic device can bring clinical benefits assuming that through this the onset of ventilation is faster.

The primary outcome is the chest compression fraction during the first two minutes of cardiopulmonary resuscitation and secondary outcomes is chest compression fraction, compressions and ventilations quality, time to first shock and to first ventilation, user satisfaction, and providers’ self-assessed cognitive load.

The main limitations of the study is represented by the size of the advantage. The advantage in terms of primary endpoint has a logical justification that may require the use of an i-gel device compared to standard CPR. Probably yes. But there are no data useful for estimating sample size.  

The limitation represented by the simulated model with manikin is also considered by the authors.

If there is a possibility of extrapolating the results of the simulated study to reality, these aspects should be considered:

1) The context of the scenario is not considered. The simulation is carried out without external "pressure" (witnesses, relatives, law enforcement, rain if carried out outdoors and so on) which in reality requires a specific management effort by the rescuers that can affect the outcomes.

2) The psychological aspect of the simulation which generally determines lower concentration than reality; this in trained people can be a determining factor.

Reviewer 2 Report

I appreciate the opportunity to review this manuscript. The revised topic is very interesting.

The standardized use of the i-gel® device could improve the management of cardiopulmonary resuscitation.

SPIRIT check list included indicates that the study meets the requirements foreseen for its development, such as the registration number and the ethics committee among others.

The methodology is very detailed.

I recommend reviewing the bibliography. (example no 33).

One last question. This trial will be carried out in a controlled environment. How can the results be applied to the real environment?

Kind regards.
